# (-)-Englerin-A Has Analgesic and Anti-Inflammatory Effects Independent of TRPC4 and 5

**DOI:** 10.3390/ijms22126380

**Published:** 2021-06-15

**Authors:** João de Sousa Valente, Khadija M Alawi, Sabah Bharde, Ali A. Zarban, Xenia Kodji, Dibesh Thapa, Fulye Argunhan, Brentton Barrett, Istvan Nagy, Susan D. Brain

**Affiliations:** 1Section of Vascular Biology and Inflammation, BHF Cardiovascular Centre of Research Excellence, School of Cardiovascular Medicine and Sciences, King’s College London, Franklin-Wilkins Building, London SE1 9NH, UK; khadija.m.alawi@gmail.com (K.M.A.); s.bharde@qmul.ac.uk (S.B.); ali.zarban@kcl.ac.uk (A.A.Z.); xenia.m.kodji@kcl.ac.uk (X.K.); dibesh.thapa@kcl.ac.uk (D.T.); fulye.argunhan@kcl.ac.uk (F.A.); brentton.barrett@kcl.ac.uk (B.B.); sue.brain@kcl.ac.uk (S.D.B.); 2Department of Pharmacological Sciences, Faculty of Pharmacy, Jazan University, Jazan 45142, Saudi Arabia; 3Nociception Group, Section of Anaesthetic, Pain Medicine and Intensive Care, Department of Surgery and Cancer, Imperial College London, London SW7 2AZ, UK; i.nagy@imperial.ac.uk

**Keywords:** (-)-Englerin A, TRPC5, TRPC4, carrageenan, inflammation, synovitis

## Abstract

Recently, we found that the deletion of TRPC5 leads to increased inflammation and pain-related behaviour in two animal models of arthritis. (-)-Englerin A (EA), an extract from the East African plant *Phyllanthus engleri* has been identified as a TRPC4/5 agonist. Here, we studied whether or not EA has any anti-inflammatory and analgesic properties via TRPC4/5 in the carrageenan model of inflammation. We found that EA treatment in CD1 mice inhibited thermal hyperalgesia and mechanical allodynia in a dose-dependent manner. Furthermore, EA significantly reduced the volume of carrageenan-induced paw oedema and the mass of the treated paws. Additionally, in dorsal root ganglion (DRG) neurons cultured from WT 129S1/SvIm mice, EA induced a dose-dependent cobalt uptake that was surprisingly preserved in cultured DRG neurons from 129S1/SvIm TRPC5 KO mice. Likewise, EA-induced anti-inflammatory and analgesic effects were preserved in the carrageenan model in animals lacking TRPC5 expression or in mice treated with TRPC4/5 antagonist ML204.This study demonstrates that while EA activates a sub-population of DRG neurons, it induces a novel TRPC4/5-independent analgesic and anti-inflammatory effect in vivo. Future studies are needed to elucidate the molecular and cellular mechanisms underlying EA’s anti-inflammatory and analgesic effects.

## 1. Introduction

Transient Receptor Potential Canonical 5 (TRPC5) is a cation channel that responds to a variety of activators [1], the exact nature of which remains unclear. TRPC5 was first characterised in the central nervous system and peripheral sensory nerves, where it forms channels either alone or in combination with TRPC1 and TRPC4 [1,2]. Later, TRPC5 expression was also found in joint synoviocytes where the stimulation of TRPC5 by the endogenous agonist, thioredoxin, resulted in a suppression of matrix metalloproteinases (MMPs) secretion in both human and animal synoviocytes [3]. We have previously described a role for TRPC5 in an endogenous anti-inflammatory/analgesic pathway in a model of unilateral arthritis [4], and more recently, in a model of osteoarthritis (OA) [5]. In both studies, animals lacking TRPC5 showed increased localised inflammation in the synovium characterised by the increased cellular infiltration and increased expression of pro-inflammatory mediators and extracellular remodelling enzymes, as well as enhanced synovial vascularization [4,5]. Interestingly, TRPC5 expression is reduced in human synovial samples of osteoarthritis patients, compared to health post-mortem donor [4]. Together, these results suggest that TRPC5 may by associated with an endogenous anti-inflammatory/analgesic pathway in inflammatory joint conditions.

Potent and selective pharmacological agents for TRPC5 channels have been difficult to develop because of the structural similarities with other TRPC channels, namely TRPC4 [6]. The most potent and selective activator is (-)-Englerin A (EA), a guaiane sesquiterpene which is extracted from an East African plant *Phyllanthus engleri* [7]. EA is an important tool for TRPC4/5 channel studies because it enables robust and clear activation in HEK cells transfected with either TRPC4 or TRPC5, possessing comparable EC50 values [8].

In this study, we tested the anti-inflammatory and analgesic potential of EA, in vivo, in the carrageenan model, a well-known model of acute inflammation [9]. Carrageenan is a seaweed extract, and its injection in the hind-paw results in pronounced oedema, and is accompanied by the rapid development of nociceptive behaviour [10]. This assay is very responsive to anti-inflammatory agents and has been essential for the development of drugs such as indomethacin and celecoxib [11,12].

## 2. Results

### 2.1. (-)-Englerin A (EA) Can Partially Prevent Carrageenan-Induced Inflammation and Pain Phenotype

In order to test the potential anti-inflammatory and analgesic properties of EA, we chose the in vivo carrageenan model, which induces a fast, acute and well established inflammatory response [9]. As expected, carrageenan injection in the footpad of CD1 mice resulted in a paw diameter increase, as early as 2 h post carrageenan injection, peaking at 4 h (Figure 1A). First, we carried out a pre-treatment, prophylaxis-type study. The paw diameters of mice pre-treated, 30 min prior to carrageenan injection, with EA, at the doses tested (2 mg/kg and 4 mg/kg), although still significantly higher than baseline at 2 and 4 h, were lower than the paw diameters of vehicle pre-treated mice. Pre-treatment with 4 mg/kg EA significantly lowered the paw diameter at 2 and 4 h post-carrageenan injection, when compared to vehicle pre-treated mice.

Moreover, 4 h after carrageenan injection, hindpaws were severed by the ankle and their masses were measured. In all experimental groups, the masses of the ipsilateral paws were significantly higher than contralateral paws (Figure 1B). However, the mass of the ipsilateral paws of the animals pre-treated with EA with 2 and 4 mg/kg were significantly lower than the mass of the ipsilateral paws of the animals pre-treated with vehicle.

To further characterize the impact of EA on inflammation, which is a cardinal feature of this model, we measured neutrophil accumulation by myeloperoxidase (MPO) assay (an enzyme primarily localised to neutrophils). This assay provides information on the recruitment of neutrophils at the inflammatory site. Here, we found that the MPO levels detected in the paws of mice pre-treated with EA at 4 mg/kg were significantly reduced compared to vehicle-pre-treated carrageenan treated paws (Figure 1C). Collectively, these results indicate that pre-treatment with EA reduced the magnitude of the oedema and inflammation induced by carrageenan.

In addition to the inflammatory outcomes, we also tested the effect of EA pre-treatment on the nociceptive behaviour induced by carrageenan injection. In accordance with the observed anti-inflammatory effect, carrageenan injection resulted in the progressive reduction of thermal thresholds (Figure 1D, Appendix A) and mechanical thresholds (Figure 1E, Appendix A), that became significantly different from baseline values at 1 or 2 h post-injection, respectively, but peaked at 3 and 4 h post-injection. EA pre-treatment reduced both thermal and mechanical hyperalgesia in carrageenan injected paws in a dose-dependent manner, reaching statistical significance only at the 4 mg/kg dose. However, interestingly, the inhibitory effect on thermal thresholds was lost at 3 h post injections.

In summary, EA reduced the oedema, neutrophil accumulation and pain-related behaviour induced by carrageenan injection. The effect was most prominent at the higher 4 mg/kg dose, and thus, we utilised this for the remainder of the studies.

### 2.2. EA Can Reverse Carrageenan-Induced Inflammation and Pain Phenotype

We also studied the ability of EA (2 mg/kg) given after the onset of carrageenan-induced inflammation, to reverse carrageenan-induced hyperalgesia, and paw oedema, in CD1 mice. EA was administered 2 h post-induction of paw inflammation and we observed that EA administration significantly reversed the carrageenan paw oedema (Figure 2A), paw mass (Figure 2B) and thermal thresholds (Figure 2C, Appendix A).

### 2.3. EA Effects on Carrageenan-Induced Inflammation and Pain-Related Behaviour Are Not Mediated by TRPC5

EA was shown to activate TRPC5-transfected HEK cells [8]. To test if EA effects on carrageenan-induced oedema formation and inflammation was mediated by TRPC5 channels, we utilized mice with the TRPC5 knockout out, on a background 129 S1/Svlm. We pre-treated 129 S1/Svlm TRPC5 WT and KO mice with 4 mg/kg EA, 30 min before carrageenan injection in the hindpaw (similar to Figure 2). Carrageenan injection in the footpad resulted in a paw diameter increase, as early as 2 h post-carrageenan injection, peaking at 4 h in both WT mice and TRPC5 KO mice (Figure 3A). EA pre-treatment reduced the paw diameter of WT mice, as observed above. Surprisingly, in TRPC5 KO mice, EA still reduced the carrageenan-induced oedema (Figure 3A). A similar effect was observed at 4 h, when paws from all experimental groups were weighed (Figure 3B). Carrageenan injection resulted in a significant increase in the mass of ipsilateral paws, both in WT and TRPC5 KO mice. EA pre-treatment significantly reduced the mass of the WT and TRPC5 KO ipsilateral paws compared to respective vehicle-treated controls (Figure 3B). In accordance, the MPO assay showed that the MPO levels were reduced after EA pre-treatment, both in WT and TRPC5 KO mice (Figure 3C).

We also looked at differences in nociceptive behaviour induced by carrageenan injection following EA pretreatment, between WT and TRPC5 KO mice. As observed above, carrageenan injection resulted in the progressive decrease of thermal thresholds (Figure 3D, Appendix A) and mechanical thresholds (Figure 3E, Appendix A), that became significantly different from baseline at 1 or 2 h respectively, but peaked at 3 and 4 h, in both WT and TRPC5 KO mice. EA pre-treatment reduced both thermal and mechanical hyperalgesia in carrageenan-injected paws in both WT and TRPC5 KO mice.

In summary, these data collectively show that the in vivo anti-inflammatory and analgesic effects of EA in the carrageenan model are not mediated by TRPC5.

### 2.4. EA Induces TRPC5 Independent Cobalt Uptake, in Cultured Dorsal Root Ganglia (DRG) Neurons

To further support our in vivo findings, next we studied the effect of EA on TRPC5-mediated responses in primary sensory neurons in culture. Non-selective cationic channels such as TRPC5 are also permeable to cobalt (Co^2+^) ions [13]. It was previously shown in TRPC5-transfected cells in culture that EA could induce the calcium uptake, in a dose-dependent manner [8]. We have previously demonstrated that dorsal root ganglia express TRPC5 channels [5]. Here, we quantified the accumulation of cobalt in cultured DRG neurons induced by EA stimulation. In this technique, the activation of channels (for example, via ligand application) in the presence of extracellular Co^2+^ leads to the intracellular accumulation of Co^2+^ ions which, when reacting with ammonium sulphide, leads to the formation of a dark brown precipitate.

In control experiments, the incubation of 129 S1/Svlm WT DRG neurons with the TRPV1 agonist capsaicin (1µM) for 5 min in the presence of CoCl_2_ induced Co^2+^ influx in 57.4 ± 8.4% of cells (*n* = 810) (Figure 4A). Likewise, in cultured DRG neurons from TRPC5 KO mice, capsaicin resulted in CO^2+^ influx in 44.08 ± 10.8% of cells (*n* = 558). There was no statistical difference between the percentage of capsaicin responders in cultured DRG neurons from WT and TRPC5 KO mice.

Incubation in CoCl_2_ buffer containing increasing concentrations of EA also induced Co^2^^+^ influx (Figure 4A) in cultured DRG neurons from WT mice. EA administration in cultured DRG neurons from TRPC5 KO mice produced a similar dose-response curve (Figure 4A, Appendix A). The concentration of EA required for 50% activation (EC50) was 8.9 ± 2.05 nM in WT DRG neurons and 10.4 ± 1.46 nM in TRPC5 KO mice. There was no statistical difference between the EC50 in WT and TRPC5 KO mice (*p* = 0.57, Student’s t test) (Figure 4C).

To further examine possible differences between the EA responders in cultured WT DRG neurons and responders from TRPC5 KO DRG neuronal cultures, we compared the size distribution of both populations (Figure 4C). The average size of EA-responders in cultures from WT DRG neurons was 1043 ± 123 µm^2^ and the size of EA-responders in TRPC5 KO cultures was 945 ± 217 µm^2^) There was no statistical difference between the sizes of both populations (*p* = 0.67, Student’s test). In conclusion, our results indicate that both in vivo and in vitro effects of EA are not mediated by TRPC5.

### 2.5. Blocking TRPC4/5 Does Not Prevent EA Effects in the Carrageenan Model

EA has been shown to activate TRPC4 channels and we have found that DRG neurons express TRPC4 channels [5]; thus, we hypothesised that the TRPC4 channel could be responsible for mediating EA effects in the carrageenan model. To test this hypothesis, we pre-treated 129 S1/Svlm WT mice with ML204 (4 mg/kg, *i.p.*), a TRPC4/5 antagonist with a half-life of 2 h, 30 min, before pre-treating mice with EA (4 mg/kg) and then injected mice with carrageenan.

As previously shown, carrageenan injection in the footpad resulted in a paw diameter increase, as early as 2 h post carrageenan injection, peaking at 4 h (Figure 5A). EA pre-treatment reduced the paw diameter of WT mice pre-treated, as observed above. In mice treated with ML204, EA still reduced the carrageenan-induced oedema (Figure 5A). A similar effect was observed at 4 h, when the experiment was terminated and paws from all experimental groups were weighed (Figure 5B). ML204 pre-treatment failed to prevent the paw mass lowering effect of EA in the ipsilateral paws compared to respective vehicle treated controls (Figure 5B). In accordance, the MPO assay showed that the MPO levels, both in vehicle + EA pre-treated and in ML204 + EA pre-treated groups, EA pre-treatment reduced MPO levels (Figure 5C).

When we determined differences in nociceptive behaviour induced by carrageenan injection following ML204 pre-treatment, we found that ML204 failed to prevent the EA-induced reduction of both thermal (Figure 5D, Appendix A) and mechanical hyperalgesia (Figure 5, Appendix A) in carrageenan-injected paws. Collectively, these data demonstrate that EA anti-inflammatory and analgesic actions in the carrageenan model are unlikely to be mediated by either TRPC4 or TRPC5.

## 3. Discussion

In this study, we have described, for the first time, an in vivo anti-inflammatory and analgesic effect of (-)-Englerin-A (EA), in a model of murine acute inflammation. Using the carrageenan model, we observed that EA inhibited thermal hyperalgesia and mechanical allodynia in a dose-dependent manner. Moreover, EA significantly reduced the volume of carrageenan-induced paw oedema and the mass of the treated paws and reduced neutrophil accumulation. We were able to show these effects with pre-treatment (prophylactic approach) and treatment when the carrageenan response had developed (therapeutic approach). Additionally, we used two strains of mice, as well as TRPC5 gene-deleted mice. Furthermore, we also demonstrate, for the first time, that EA activates a population of cultured primary sensory neurons. Overall, we have confirmed that both the analgesic and anti-inflammatory effects of EA in vivo, and the activation of DRG neurons in vitro, are not mediated by TRPC5 channels.

We set out to use EA as a tool to characterise the contribution of TRPC5 receptor in modulating acute inflammation. EA is a well-established agonist of TRPC4 and TRPC5 non-selective cation channels [8,14]. We previously described a role for TRPC5 in an endogenous anti-inflammatory/analgesic pathway in a model of unilateral arthritis [4], and more recently, in a model of osteoarthritis [5]. In both studies, animals lacking the expression of TRPC5 showed increased localised inflammation in the synovium characterised by the increased cellular infiltration and increased expression of pro-inflammatory mediators and extracellular remodelling enzymes, as well as enhanced synovial vascularization [4,5].

Here, we used the carrageenan model of inflammation, which is an acute, well characterised and highly reproducible model of inflammation [9]. When injected into the paws, carrageenan immediately induces clear signs of oedema, hyperalgesia and erythema. While the model is highly dependent on proinflammatory mediators such as bradykinin, histamine, reactive oxygen and nitrogen species being produced and released at the site of injection, it is also well established that neutrophils rapidly migrate to the site of inflammation in this model [10]. This model has also been important in drug development [12], and in our study, EA significantly alleviated the oedema caused by carrageenan in CD1 mice, in a dose-dependent manner. The effect was present, both when EA was administered before the onset of inflammation, and two hours after carrageenan injection. Currently, there are only a few reports where EA was administered in vivo, but none have investigated analgesic or anti-inflammatory effects. The scarcity of the literature could be partially explained by the cytotoxic effect of EA, particularly at higher doses. However, in our studies, we did not observe any toxicity issues with EA injection at both 2 mg/kg and 4 mg/kg, but some mice did display a disorientated appearance, such as the apparent reduction of locomotor activity. Importantly, these were acute effects and the mice returned to their normal appearance soon after administration. A recent study demonstrated that such adverse reactions are dependent on both TRPC4 and TRPC5 [15]. Furthermore, the study demonstrated that when given intraperitoneally, EA is peripherally restricted, and hence, the adverse reaction is likely to be mediated peripherally [15].

We show that EA reduced thermal and mechanical hyperalgesia in the carrageenan model. This was in accordance with previous data from this laboratory that showed that TRPC5 expression had a protective role in inflammation- and pain-related behaviour, secondary to the development of experimental arthritis [4,5]. To examine the hypothesis that TRPC5 was mediating the EA effect in vivo, we tested TRPC5 KO mice in the carrageenan model and observed that neither the anti-inflammatory nor the analgesic effects were mediated through TRPC5. To further confirm this, we also tested the ability of EA to induce cobalt uptake in cultured DRG neurons, which we have previously expressed in TRPC5 channels [5]. In these cultured cells, EA induced cobalt influx in a dose dependent manner, similar to what was observed in TRPC5-transfected HEK cells [8], however, in cultured DRG neurons from TRPC5 KO mice, the EA activity was surprisingly still present. Together, our data demonstrate that both in vivo and in vitro effects of EA are not mediated solely by TRPC5 channels. One possible candidate mediating cobalt uptake in cultured DRG is TRPC4, which we have found, previously, to be expressed in DRG neurons [5]. Furthermore, selectivity assays have demonstrated that EA can bind other TRP channels which are also expressed in DRG neurons [14,16]. Future in vitro studies employing pharmacological tools or cells from transgenic animals could help elucidate which channel/s mediate the observed cobalt uptake in DRG.

The screening of plant extracts in the search of new anti-cancer agents resulted in the discovery of EA as an agent with potent cytotoxicity against renal cancer cells and a small subset of other cancer cells [7,17]. There have been multiple attempts to identify the molecular targets of EA, and several have been proposed [14]. One study found that EA at a dose of 5 mg/kg reduced xenograft tumour growth in athymic mice via PKCθ stimulation [18]. More recently, studies suggested that its TRPC4/5 activation by EA that inhibits tumour cell proliferation [14] and that EA achieves cancer cell cytotoxicity by inducing sustained Na^+^ entry through heteromeric TRPC1/TRPC4 channels [19]. There are currently no inhibitors capable of differentiating between TRPC4 and 5 [6]. The best characterised inhibitor is the TRPC4/5 inhibitor ML204 [20], which inhibits TRPC4 with an IC_50_ of about 1 μM and caused about a 65% inhibition of TRPC5 at 10 μM [20]. To test if the anti-inflammatory and analgesic effects of EA in the carrageenan model were mediated by TRPC4, we pre-treated mice with ML204 but, to our surprise, we found that neither of the anti-inflammatory nor the analgesic effects of EA were blocked. These results could be partially explained by the relative resistance of the TRPC4-TRPC1 heteromer to ML204 [21]. Indeed, previous results show that DRG neurons express both TRPC1 and TRPC4 channels [5]. Another possible explanation is that EA acts through a non-TRPC4/5 receptor in a less specific manner. Indeed, EA has been shown to weakly inhibit TRPA1, TRPV3/V4, and TRPM8 [14] receptors, all of which are generally accepted to play a role in pain processing [22,23,24,25].

In conclusion, we demonstrate, for the first time, that EA possesses anti-inflammatory and analgesic properties in the carrageenan model of acute inflammation. However, the lack of an identified target, along with the reported cytotoxicity of EA, suggests that this compound may not be valuable for drug discovery, but it may, nonetheless, be useful in helping to identify the pathways involved in pain and inflammation.

## 4. Materials and Methods

### 4.1. Animals

A total of 49 male CD1 mice (20 g) and 58 129 S1/SvIm wild type (WT) and TRPC5 knock out (KO) adult male mice (3–5 months, 25–35 g), previously described (Alawi et al., 2017), were used in this study. Food and water were available ad libitum and mice were housed under standard conditions with a 12-h light/dark cycle. Experiments were performed by investigators blinded to the identity of the animals. Animals were designated experimental numbers, in a randomised manner, by an uninformed independent researcher, who secured and concealed the allocation until the end of the study. All procedures were performed according to the UK Animals (Scientific Procedures) Act 1986 and approved by the King’s College London Animal Care and Ethics Committees. Furthermore, we adhere to Good Laboratory Practice following the ARRIVE guidelines [26], and the principles of the 3Rs (Replacement, Reduction and Refinement). Behavioural studies were conducted every hour for 4 h, after which the study was terminated, and tissues were collected for further analysis.

### 4.2. Carrageenan-Induced Paw Oedema

Adult male mice were anaesthetized by 2% isoflurane carried in 0.5 L/min O_2_ and paw oedema was induced by injecting 50 µl of 2% carrageenan (100 mg, 5 mL saline) into the pad region of the glabrous skin on the underside of the left hind paw of male mice using a 25 G needle.

### 4.3. Drugs

(-)-Englerin A (EA) (2 or 4 mg/kg; ITW Reagents, Germany) or vehicle (0.5% EtOH, 1% PEG300, 0.5% Cremaphor EL in saline, *i.p.*) was given 30 min before carrageenan injection. In some experiments, mice were treated with the TRPC4/5 antagonist ML204 (2 mg/kg; Tocris, Bristol, U.K.) or vehicle (2% DMSO in saline, *i.p.*) 1 h prior to the induction of paw oedema.

### 4.4. Behavioural Assays

Mechanical withdrawal thresholds were assessed by applying calibrated von Frey monofilaments (0.07–1.00 g, North Coast Medical, Morgan Hill, USA) to the plantar surface of the hind paw to unrestrained mice, placed in individual Plexiglas cubicles, as described previously [27]. The 50% paw withdrawal threshold (PWT) was determined by increasing or decreasing stimulus intensity and estimated using the Dixon “up–down” method [28] at 2 and 4 h post-carrageenan injection to prevent overlap with thermal testing.

Changes to thermal thresholds were assessed by the Hargreaves test (Hargreaves et al. 1988), 1 and 3 h post carrageenan injection, as described previously [27]. Briefly, mice were placed in a Perspex box and a thermal stimulus from a radiant heat source (a high-intensity projector lamp bulb), which is able to deliver a constantly increasing thermal stimulus, was directed to the plantar surface of the paw (Ugo Basile, Gemonio, Italy). The time until the animal voluntarily withdrew the paw was measured.

### 4.5. Joint Thickness Measurements

Prior to the induction of paw oedema and at 2 and 4 h post carrageenan injection, animals were loosely restrained, and the paw diameter was measured using a thickness gauge (Mitutoyo, Andover, UK). Data are expressed in millimetres (mm).

### 4.6. Paw Mass Measurements

At the end of studies, animals were terminally anaesthetized with an overdose of isoflurane and death was confirmed by cervical dislocation. The paw was collected by cutting at the ankle joint, and mass in mg was measured, after which, paw samples were snap frozen in liquid nitrogen and stored at −80 °C for myeloperoxidase (MPO) assay analysis.

### 4.7. MPO Assay

Paw samples were homogenised in 1 ml of homogenisation buffer (0.1 M NaCl, 0.02 M NaPO4, 0.015 M EDTA, 0.5% hexadecyltrimethylammoniumbromide pH 4.7) in tissue homogenizer (Janke & Kunkel, IKA Works, Staufen. Germany) after which samples were centrifuged at 17,000 g for 15 min at 4 °C, and the supernatant was collected and diluted 10 times with a homogenisation buffer. Reactions were performed in 96-well plate at room temperature. H_2_O_2_ oxidation of 3,3′,5,5′-Tetramethylbenzidine Liquid Substrate System, Sigma (TMB) was used to determine MPO activity. In a 96-well plate, 25 μl of MPO buffer was added to 25 μl of each sample and a 100 μl of TMB liquid substrate was then added to each well. The plate was incubated in the dark at 37 °C for 15 min. Absorbance (OD) was read at 620 nm using a spectrophotometer and a standard curve was plotted of OD against MPO in the standard samples.

### 4.8. Dorsal Root Ganglia Cultures

DRG neurons’ cultures were prepared by collecting DRGs from the first cervical to the sixth lumbar segment into Ham’s nutrient F12 culture medium (Sigma) supplemented with 2% Ultroser G (Pall SA, Saint-Germain-en-Laye, France), 1 mM glutamine (Invitrogen, Waltham, USA), 50 IU/ml penicillin (Invitrogen, Waltham, USA) and 50 μg/ml streptomycin (Invitrogen, Waltham, USA). Following incubation in 2000 U/ml collagenase type IV (Worthington Biochemical Corp., Lakewood, USA) for 3 h, DRG were triturated, and the cells were plated onto poly-DL-ornithine (Sigma)-coated glass coverslips. Cells were grown for 24 h, at 37 °C in the supplemented medium, to which the nerve growth factor (NGF, 50 ng/mL; Promega, Southampton, UK) was added.

### 4.9. Cobalt Uptake

The (-)-Englerin-(A) activation of cultured DRG neurons was assessed by cobalt uptake, as described previously [29]. Coverslips were washed for 2 min in buffer solution (in mM: 57.5 NaCl, 5 KCl, 2 MgCl_2_, 10 HEPES, 12 glucose, and 139 sucrose, pH 7.4), and then cells were incubated in the buffer solution containing 5 mM CoCl_2_ (cobalt-uptake solution) and increasing concentrations of EA (in nM: 0.3, 1, 3, 10, 30 or 100). Negative and positive control experiments were performed for each culture, in which control buffer or 1 µM capsaicin was added to the cobalt-containing buffer, respectively. Cobalt taken up by the neurons then was precipitated with 2.5% 2-mercaptoethanol solution in buffer for 1 min. Cells were fixed in 70% ethanol. The cobalt precipitate was visualized by light microscopy using a Leitz Diaplan microscope (Leica, Weltzar, Germany). Cells were identified as labelled and non-labelled cells by a researcher blinded to the experiments, and the cytoplasm was marked as region of interest (ROI) using ImageJ (NIH). The area and mean pixel intensity of the ROIs were then measured. At least 100 cells per coverslip were analysed.

### 4.10. Data Analysis

Animals were randomised to experimental groups by an external researcher, as described above. Data were tested for normal distribution and analysed using GraphPad Prism 8.0 (GraphPad, San Diego, CA, USA). The number of animals used in the various experiments is reported in the figure legends. Data analysis for in vivo assays was performed by two-way repeated measures ANOVA, followed by Tukey’s post hoc analysis, with the factors considered being time and treatment/genotype. MPO assay data were analysed by two-way ANOVA, followed by Tukey’s test, with factors considered being treatment and genotype. The statistical tests performed, and the numbers of animals used, are displayed in the figure legends. All data are presented as mean ± S.E.M. In all analyses, *p* < 0.05 was taken to indicate statistical significance.

## Figures and Tables

**Figure 1 ijms-22-06380-f001:**
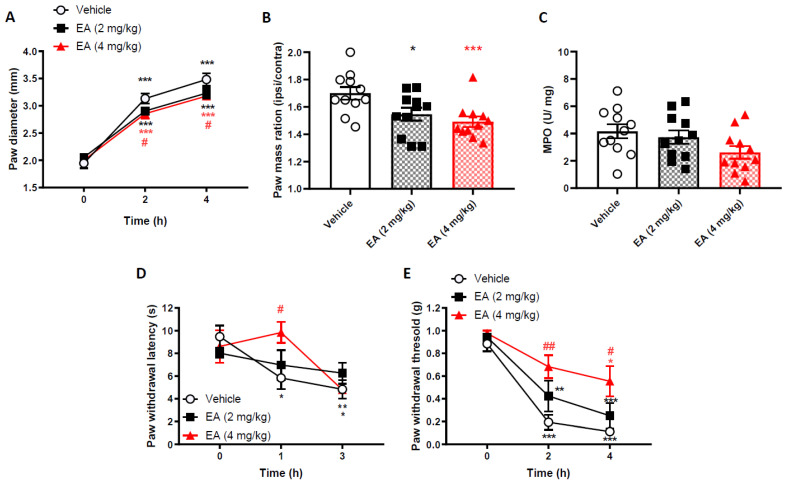
Englerin-A given as a pretreatment reduces carrageenan-induced paw oedema and mechanical and thermal hyperalgesia, in a dose-dependent manner. (**A**) Time-course of paw oedema diameter measured by callipers and (**B**) ipsilateral/ contralateral paw mass ratio 4 h following induction of paw oedema 2% Carrageenan. Data are mean ± S.E.M, *n* = 11. * *p* < 0.05, *** *p* < 0.001 versus baseline, # *p* < 0.05, versus vehicle, Two-way Repeated measures (RM) ANOVA, *post-hoc* Tukey test. (**C**) Myeloperoxidase (MPO) levels in paw exudates, 4 h after the induction of paw inflammation. Data are presented as mean ± S.E.M, *n* = 11. (**D**) Thermal and (**E**) mechanical paw withdrawal thresholds measured at baseline and at 1 and 3 or 2 and 4 h, respectively, after intra-plantar injections of carrageenan in the left hindpaw. * *p* < 0.05, ** *p* < 0.01, *** *p* < 0.001 versus baseline, # *p* < 0.05, ## *p* < 0.01 versus vehicle, Two-way RM ANOVA, post hoc Tukey test. Data are mean ± S.E.M, *n* = 8 (Hargreaves test: *n* = 6).

**Figure 2 ijms-22-06380-f002:**
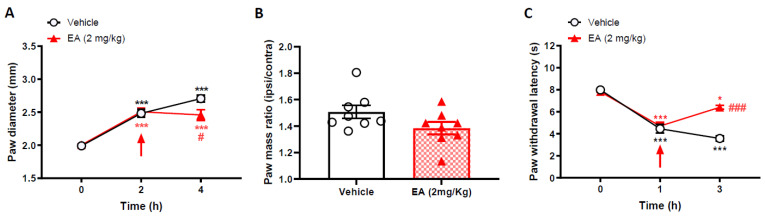
Englerin-A reverses carrageenan-induced paw oedema and thermal hyperalgesia when given 2 h after carrageenan. (**A**) Time-course of paw oedema measured by callipers and (**B**) ipsilateral/ contralateral paw mass ratio, 4 h following induction of paw oedema by 2% Carrageenan. Data are mean ± S.E.M, *n* = 6–8. *** *p* < 0.001 versus baseline, # *p* < 0.05, versus vehicle, Two-way RM ANOVA, post-hoc Tukey test. (**C**) Thermal paw withdrawal thresholds measured at 1 and 3 h, respectively, after intra-plantar injections of carrageenan. Arrow indicates time of EA treatment (2 mg/kg i.p.). * *p* < 0.05, *** *p* < 0.001 versus baseline, ### *p* < 0.001 versus vehicle, Two-way ANOVA, post-hoc Tukey test. Data are mean ± S.E.M, *n* = 6–8.

**Figure 3 ijms-22-06380-f003:**
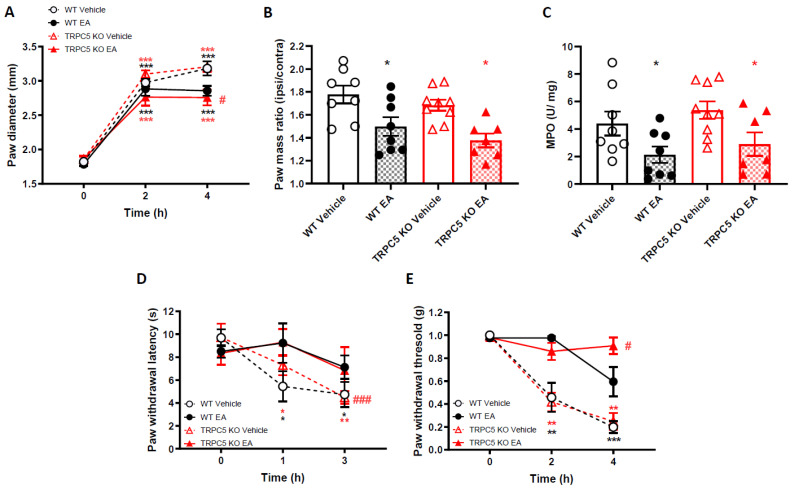
Englerin-A reduces carrageenan-induced paw oedema and mechanical and thermal hyperalgesia, in both WT and TRPC5 KO mice. (**A**) Time-course of paw oedema measured by callipers and (**B**) ipsilateral/contralateral paw mass ratio 4 h following induction of paw oedema by 2% Carrageenan. Data are mean ± S.E.M, *n* = 8. * *p* < 0.05, *** *p* < 0.001 versus baseline, # *p* < 0.05, versus vehicle, Two-way ANOVA, post hoc Tukey test. (**C**) Myeloperoxidase (MPO) levels in paw exudates, 4 h after the induction of paw inflammation. Data are mean ± S.E.M, *n* = 8. * *p* < 0.05 versus vehicle, unpaired t-test. (**D**) Thermal and (**E**) mechanical paw withdrawal thresholds measured at baseline and at 1 and 3 or 2 and 4 h, respectively, after intra-plantar injections of carrageenan. * *p* < 0.05, ** *p* < 0.01, *** *p* < 0.001 versus baseline, # *p* < 0.05, ### *p* < 0.001 versus vehicle, Two-way RM ANOVA, post hoc Tukey test. Data are mean ± S.E.M, *n* = 8.

**Figure 4 ijms-22-06380-f004:**
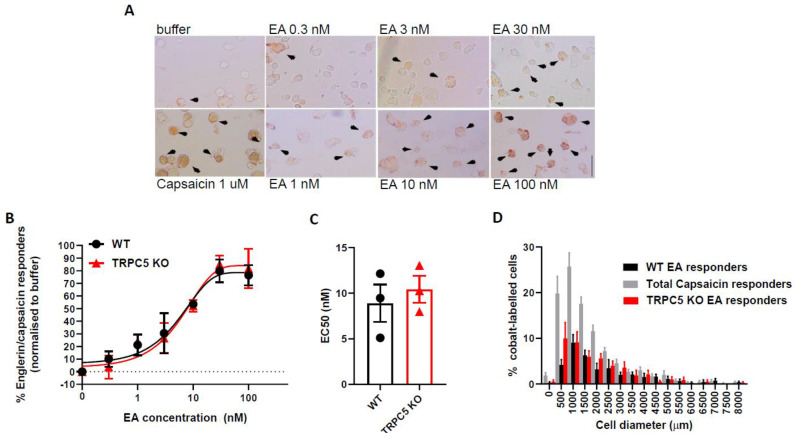
Englerin-A induces cobalt uptake in cultured primary sensory neurons. (**A**) Representative images of cobalt-labelled primary sensory neurons (brown; indicated by black arrowheads) obtained from dorsal root ganglia from WT or TRPC5 −/− mice, stimulated with ascending concentrations of Englerin-A (scale bar 200 µm; magnification 10×). Capsaicin (1µM) and buffer alone were used as positive and negative controls respectively. (**B**) Dose-response curve of cultured primary sensory neurons from WT (black; *n* = 3) and KO (red, *n* = 3) mice to Englerin-A (**C**) Comparison between EC50 for Englerin A in WT DRG cultures (black) and TRPC5 KO DRG cultures (red). (**D**) Size-frequency distribution of cobalt-labelled DRG neurons from WT (dark grey, *n* = 690) and KO mice (red, *n* = 449) to Englerin-A (30 nM) stimulation and total capsaicin responsive neurons (light grey, *n* = 1453).

**Figure 5 ijms-22-06380-f005:**
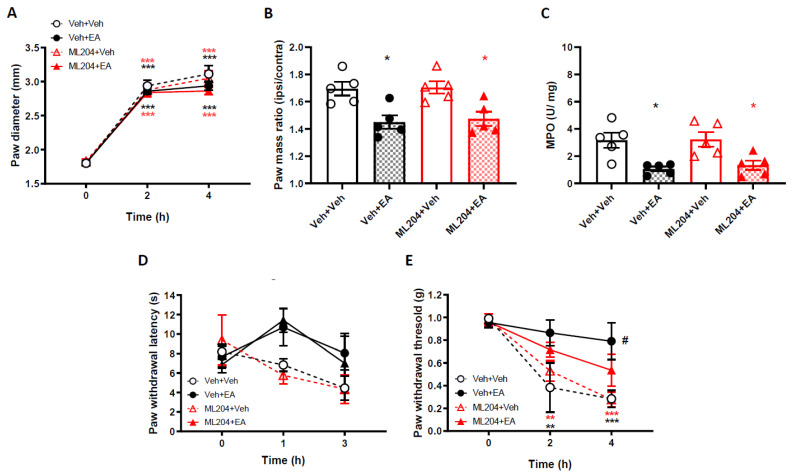
ML204 fails to revert EA effect on carrageenan-induced paw oedema and mechanical and thermal hyperalgesia. (**A**) Time-course of paw oedema measured by callipers and (**B**) ipsilateral/contralateral paw mass ratio 4 h following induction of paw oedema by 2% Carrageenan. Data are mean ± S.E.M, *n* = 5. * *p* < 0.05, *** *p* < 0.001 versus baseline, versus vehicle, Two-way ANOVA, post hoc Tukey test. (**C**) Myeloperoxidase (MPO) levels in paw exudates, 4 h after the induction of paw inflammation. Data are mean ± S.E.M, *n* = 5. * *p* < 0.05 versus vehicle, Unpaired *t*-test. (**D**) Thermal and (**E**) mechanical paw withdrawal thresholds measured at baseline and at 1 and 3 or 2 and 4 h, respectively, after intra-plantar injections of carrageenan. ** *p* < 0.01, *** *p* < 0.001 versus baseline, # *p* < 0.05, versus vehicle, Two-way RM ANOVA, post hoc Tukey test. Data are mean ± S.E.M, *n* = 5.

## Data Availability

The data presented in this study are available on request from the corresponding author.

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
