# Peer review of "(-)-Englerin-A Has Analgesic and Anti-Inflammatory Effects Independent of TRPC4 and 5"

_ijms, 2021, doi:10.3390/ijms22126380_

Round 1

Reviewer 1 Report

The authors highlight the capacity of (-)-Englerin-A as an analgesic and anti-inflammatory agent independent of TRPC4/5 in a carrageenan model of inflammation. In this study, behavioural and pathological characterisation of EA on inflammatory nociception were evaluated in mice. In general the manuscript is well-written. 

Some major revisions:

1) Figure 1B: It would be better illustrated, if the authors plotted relative ipsilateral paw mass/contralateral paw mass per animal. This would enable to better visualise the trends as the EA treated animals also display a slightly lower contralateral paw mass relative to vehicle controls. Also, similar suggestion for figure 3B.

2) Authors are requested to show data for MPO evaluation as in Figure 1C when pre-treated with EA (2mg/kg)? Was the effect in a dose-dependent manner?

3) Line 86: Authors have jumped to the results on inflammation. It would be suggested that a brief intro in order to link this results to the previous data. For example: Subsequent to behavioral assay and gross pathology, we went on to evaluate the impact of EA on inflammation, a cardinal feature of this model.

4) Lines 105-109, Results section 2: Not entirely sure for the inclusion of this section and Figure 2. The authors have presented these results in Figure 1 for the paw diameter (1A), paw mass (1B), and PWL (1D). More than being further reinforcing, the authors are undermining their findings from the previous figure. In particular, Figure 1A demonstrates no significant effect of 2mg/kg EA on paw diameter at any interval; however, figure 2A is demonstrating a signification reduction in the diameter at 4 hours post carrageenan administration. Likewise, in Figure 1D the authors show no effect of 2mg/kg EA on PWL at any time interval; but figure 2C shows a significantly greater PWL in EA-treated animals 3 hours post carrageenan injection. Therefore, Figure 2 as a whole is in itself countering the findings presented in Figure 1. It raises further issues with regards to the reproducibility and robustness of the data.

5) Both Figures 3D and 3E raise concerns about the robustness of the EA effect. As in Figure 1D 4 mg/kg EA pre-treatment delayed the thermal hypersensitivity 1h post carrageenan, however this is not evident in Figure 3D. Likewise in Figure 1E, the authors demonstrate 4mg/kg EA pre-treatment delays mechanical hypersensitivity by in Figure 3D this has not been shown. I would suggest that the authors also present a table of summary as supplementary data if wished, to highlight the mean, SEM, and respective p values for comparisons.

6) Figure 4A: Addition of either a table or a graph plot would be easy to understand the observation presented in addition to images.

7) Figure 5A: Was the reduction in Veh+EA significant compared to Veh +Veh? And was this the case for the ML204 groups? Not clear from the graphs.

8) Line 295: Any pathways that can be suggested by authors for future direction for the underlying mechanisms of EA? It is clearly a missing link in the article that is the key for the novelty of this publication.

Minor revisions:

1) Provide full form (abbreviations) and be consistent in using the abbreviations once introduced. For example: osteoarthritis (line 39) but abbreviation introduced in line 44 "osteoarthritis (OA)". Example 2: Co2+ ions (line 154) provide full form followed by abbreviations in bracket.

2) Reconsider using the term "pain" in animals throughout the text. Would recommend replacing it with either pain-like behaviour or nociceptive sensitivity.

3) Throughout the text change in-vivo and in-vitro to italic font. 

4) Be consistent with "pre-treatment" as at several occasions it is written as "pretreatment" (i.e. line 143).

5) Check spellings: i.e. line 102 - "In summary EA reduced both the oedema, neutrophil acomulation..."

6) Additional grammar corrections in attachment.

Author Response

The authors highlight the capacity of (-)-Englerin-A as an analgesic and anti-inflammatory agent independent of TRPC4/5 in a carrageenan model of inflammation. In this study, behavioural and pathological characterisation of EA on inflammatory nociception were evaluated in mice. In general, the manuscript is well-written. 

Some major revisions:

  • Figure 1B: It would be better illustrated if the authors plotted relative ipsilateral paw mass/contralateral paw mass per animal. This would enable to better visualise the trends as the EA treated animals also display a slightly lower contralateral paw mass relative to vehicle controls. Also, similar suggestion for figure 3B.

Reply: Thank you for your valuable suggestion. We have modified all figures depicting the data related to paw mass and agree that it is easier to interpret. (Figures 1, 2, 3 and 5)

  • Authors are requested to show data for MPO evaluation as in Figure 1C when pre-treated with EA (2mg/kg)? Was the effect in a dose-dependent manner?

Reply: As requested we have added the data related to the MPO levels on animals treated with 2 mg/kg on Figure 1C.

  • Line 86: Authors have jumped to the results on inflammation. It would be suggested that a brief intro in order to link this results to the previous data. For example: Subsequent to behavioral assay and gross pathology, we went on to evaluate the impact of EA on inflammation, a cardinal feature of this model.

Reply: We added the following sentence to the beginning of that paragraph: “To further characterize the impact of EA on inflammation, which is a cardinal feature of this model we measured neutrophil accumulation by myeloperoxidase (MPO) assay (an enzyme primarily localised to neutrophils” (Line 87, highlighted in yellow)

  • Lines 105-109, Results section 2: Not entirely sure for the inclusion of this section and Figure 2. The authors have presented these results in Figure 1 for the paw diameter (1A), paw mass (1B), and PWL (1D). More than being further reinforcing, the authors are undermining their findings from the previous figure. In particular, Figure 1A demonstrates no significant effect of 2mg/kg EA on paw diameter at any interval; however, figure 2A is demonstrating a signification reduction in the diameter at 4 hours post carrageenan administration. Likewise, in Figure 1D the authors show no effect of 2mg/kg EA on PWL at any time interval; but figure 2C shows a significantly greater PWL in EA-treated animals 3 hours post carrageenan injection. Therefore, Figure 2 as a whole is in itself countering the findings presented in Figure 1. It raises further issues with regards to the reproducibility and robustness of the data.

Reply: The experiments in figure 1 and 2 had different experimental plans and therefore different outcomes are not surprizing. We have now attempted to explain this more clearly.  Figure 1 is a pre-treatment of EA, given 5 min before the carrageenan and a typical experimental design often used by researchers.  It is generally considered that the early phase of inflammation onset may be modulated (as shown) and this in turn can affect any later response.  Figure 2 is designed as a therapeutic treatment (more typical of the way that treatments are given clinically) Thus whilst it is hypothesised that similar results will be obtained, they cannot be directly compared, due to the differing protocols. On the other hand it is very positive that Fig 2 shows a similar inhibitory profile as Fig 1.   We do agree with the reviewer that differences exist but hopefully this further explanation now makes the reason why this is likely clear.

  • Both Figures 3D and 3E raise concerns about the robustness of the EA effect. As in Figure 1D 4 mg/kg EA pre-treatment delayed the thermal hypersensitivity 1h post carrageenan, however this is not evident in Figure 3D. Likewise in Figure 1E, the authors demonstrate 4mg/kg EA pre-treatment delays mechanical hypersensitivity by in Figure 3D this has not been shown. I would suggest that the authors also present a table of summary as supplementary data if wished, to highlight the mean, SEM, and respective p values for comparisons.

Reply: We have now improved our explanation, so that it is clear that we have used different strains of mice in these protocols (multiple instances, highlighted in yellow). We use CD1 mice as they are widely available and used with success for many years in our group. However, the TRPC5 KO mice are on a different background, which is not freely available (129/Svj). Experiments depicted in Figure 3 (and also figure 5) are conducted in 129/Svj strain (the background strain of TRPC5 KO mice). Of note, this mouse strain is a less responsive mouse strain compared to other strains (PMID: 16158908, PMID: 32872338). Nevertheless, the magnitude of change in paw mass, diameter and MPO levels argues that the differences in EA effect, show similar trends observed in both strains of mice. Collectively we show that EA is anti-hyperalgesic/anti-inflammatory using three different experimental designs and we now, as the referee suggests put these data in supplementary tables.

  • Figure 4A: Addition of either a table or a graph plot would be easy to understand the observation presented in addition to images.

Reply: We added Figure 4A to provide some reference to the data expressed by figure 4B which shows that the number of cells responding with cobalt influx (brown precipitate, arrow heads on the figure) increases with increased concentrations of EA. We now also include the data of Fig 4B also as a table in the supplementary material, in keeping with the referees suggestions.

  • Figure 5A: Was the reduction in Veh+EA significant compared to Veh +Veh? And was this the case for the ML204 groups? Not clear from the graphs.

Reply: As depicted in the figure 5A, because of the low n number (n=5), we were underpowered to observed differences between Veh+EA and Veh+Veh, in terms of the two dimensional paw diameter measurement (5A), although we were able to in Fig 5B where three dimensions (mass) was measured.    

  • Line 295: Any pathways that can be suggested by authors for future direction for the underlying mechanisms of EA? It is clearly a missing link in the article that is the key for the novelty of this publication.

Reply: Englerin A is a relatively new compound and therefore only a few studies have looked to study the molecular targets of TRPC5. Carson et al showed that in aa selectivity assay, EA weekly inhibits TRPA1, TRPV3/V4, and TRPM8 (DOI: 10.1371/journal.pone.0127498). “Another possible explanation is that EA acts through a non-TRPC4/5 receptor in a less specific manner. Indeed EA has been shown to weakly inhibit TRPA1, TRPV3/V4, and TRPM8 [14], receptors, all which are generally accepted to play a role in pain processing [21-24].” (line 299, highlighted in yellow)

Minor revisions:

  • Provide full form (abbreviations) and be consistent in using the abbreviations once introduced. For example: osteoarthritis (line 39) but abbreviation introduced in line 44 "osteoarthritis (OA)". Example 2: Co2+ions (line 154) provide full form followed by abbreviations in bracket.

Reply: Thank you for your valuable comments. These have been, accordingly, changed in the text. (Line 39, Line 161, highlighted in yellow)

  • Reconsider using the term "pain" in animals throughout the text. Would recommend replacing it with either pain-like behaviour or nociceptive sensitivity.

Reply: As suggested, the term pain was replaced by pain-related behaviour. (multiple instances)

  • Throughout the text change in-vivoand in-vitro to italic font. 

Reply: These have been changed in the text. (multiple instances)

  • Be consistent with "pre-treatment" as at several occasions it is written as "pretreatment" (i.e. line 143).

Reply: Throughout the text, pre-treatment was replaced by pretreatment. (multiple instances)

  • Check spellings: i.e. line 102 - "In summary EA reduced both the oedema, neutrophil acomulation..."

Reply: The spelling has been corrected. (line 105, highlighted in yellow)

  • Additional grammar corrections in attachment.

Reply: Thank you for this. The attachment could not be found, but after contacting the editorial board of the journal we were told that “editorial department will check the grammar and edit English of the manuscript after it is accepted “

Reviewer 2 Report

In previous studies the authors described a role for TRPC5 in an endogenous anti-inflammatory/analgesic pathway in a model of unilateral arthritis and in a model of osteoarthritis. In this study, the authors described for the first time an in vivo anti-inflammatory and analgesic effect of (-)-Englerin-A (EA), in a model of murine acute inflammation. They used the carrageenan model, a well-known model of accute inflammation.

They studied if: 1. EA can partially prevent carrageenan induced inflammation and pain phenotype; 2. EA can reverse carrageenan-induced inflammation and pain phenotype; 3. EA effects on carrageenan induced inflammation and pain are not mediated by TRPC5; 4. EA induces TRPC5 independent cobalt uptake, in cultured dorsal root ganglia neurons and 5. If blocking TRPC4/5 does not prevent EA effects in the carrageenan model. The current study confirmed that both the analgesic and anti-inflammatory effects of EA in vivo, and the activation of DRG neurons in vitro, are not mediated by TRPC5 channels.

The lack of an identified target of action, along with the reported cytotoxicity of EA suggests that this compound may not be valuable for drug discovery. It may be useful in helping to identify pathways involved in pain and inflammation. To confirm these hypotheses, future studies are needed.

The methods and discussions are clearly presented. The novelty and the significance of the results are very good. I recommend publishing this article in the International Journal of Molecular Sciences (IJMS) after Minor revisions.

Some remarks:

  1. Line 388, author contributions: you forgot to mention Istvan Nagy contribution
  2. Line 298, Animals: you wrote that the study was carried out on 33 male CD1 mice (20g) and 52 129S1/SvIm wild type (WT) and TRPC5 knock out 298 (KO) adult male mice

Line 118, at point 3: To determine: EA effects on carrageenan induced inflammation and pain are not mediated by TRPC5, the study was performed on TRPC5 WT and KO mice

To determine if Englerin A (EA) can partially prevent carrageenan induced inflammation and pain phenotype - line 60 (point 1) and if EA can reverse carrageenan-induced inflammation and pain phenotype – line 105 (point 2) what type of mice were used? Please clarify the issue by writing at these points. Reading the discussion, at line 247 I understood that you used CD1 mice.

How do you explain that you chose 33 CD1 mice and 55 TRPC5 WT and KO mice?

The numbers of animals used are displayed in the figure legends. Why is the number of mice / group so different?

Author Response

In previous studies the authors described a role for TRPC5 in an endogenous anti-inflammatory/analgesic pathway in a model of unilateral arthritis and in a model of osteoarthritis. In this study, the authors described for the first time an in vivo anti-inflammatory and analgesic effect of (-)-Englerin-A (EA), in a model of murine acute inflammation. They used the carrageenan model, a well-known model of accute inflammation.

They studied if: 1. EA can partially prevent carrageenan induced inflammation and pain phenotype; 2. EA can reverse carrageenan-induced inflammation and pain phenotype; 3. EA effects on carrageenan induced inflammation and pain are not mediated by TRPC5; 4. EA induces TRPC5 independent cobalt uptake, in cultured dorsal root ganglia neurons and 5. If blocking TRPC4/5 does not prevent EA effects in the carrageenan model. The current study confirmed that both the analgesic and anti-inflammatory effects of EA in vivo, and the activation of DRG neurons in vitro, are not mediated by TRPC5 channels.

The lack of an identified target of action, along with the reported cytotoxicity of EA suggests that this compound may not be valuable for drug discovery. It may be useful in helping to identify pathways involved in pain and inflammation. To confirm these hypotheses, future studies are needed.

The methods and discussions are clearly presented. The novelty and the significance of the results are very good. I recommend publishing this article in the International Journal of Molecular Sciences (IJMS) after Minor revisions.

Some remarks:

  1. Line 388, author contributions: you forgot to mention Istvan Nagy contribution.

Reply: Thank you for this comment. Istvan Nagy (IN) has been added. “JSV, KA and SDB designed the research, JSV, KA, SB, IN and AAZ performed experiments and collected and analysed the data. AAZ, XK, DT, BB and FA helped with the blinding. All authors participated in the interpretation of the data. JSV and SDB wrote the paper. All authors read and approved the submitted manuscript.” (line 403).

  1. Line 298, Animals: you wrote that the study was carried out on 33 male CD1 mice (20g) and 52 129S1/SvIm wild type (WT) and TRPC5 knock out 298 (KO) adult male mice
  2. Line 118, at point 3: To determine: EA effects on carrageenan induced inflammation and pain are not mediated by TRPC5, the study was performed on TRPC5 WT and KO mice

To determine if Englerin A (EA) can partially prevent carrageenan induced inflammation and pain phenotype - line 60 (point 1) and if EA can reverse carrageenan-induced inflammation and pain phenotype – line 105 (point 2) what type of mice were used? Please clarify the issue by writing at these points. Reading the discussion, at line 247 I understood that you used CD1 mice.

Reply: Thank you for your comments. Accordingly, the strain of the mice were added at the beginning of the description of the results, in the results section to clarify which strains used in each experiment. (lines 64, 110, 127, 169 and 201).

How do you explain that you chose 33 CD1 mice and 55 TRPC5 WT and KO mice?

Reply: 33 refer to the number of mice in figure 1 and 52 refer to the number of mice used in figure 3 and 5. We apologise but by mistake, the animals from Figure 2 were not added neither the ones used in figure 4. Therefore the count is 49 CD1 and 58 129S1/SvIm mice. This has been corrected in the materials and methods section (line 310).

The numbers of animals used are displayed in the figure legends. Why is the number of mice / group so different?

Reply: The number of mice per experimental group was kept consistent within an experiment with similar numbers between experimental groups. Between experiments the total number of animals used was constrained by the number of WT and TRPC5 KO littermates available of a similar age and size.

Reviewer 3 Report

The article titled “(-)-Englerin-A has analgesic and anti-inflammatory effects in-2 dependent of TRPC4 and 5” further deepens the knowledge on this compound's anti-inflammatory and analgesic mechanism of action. The research was well planned, and the methods are well described.

In order to improve the manuscript, the authors should address some critical points, mainly in the results section.

  1. English language revision and spell check are recommended (e.g. “in vitro” in italic; in the abstract, EA is a compound extracted from a plant, not an extract, please revise; TRPC description, in the literature, is found “transient receptor potential cation channel”; Avoid the use of the same expression repeatedly, e.g. “a role for TRPC5 in an endogenous (…) in a model (…) in a model of osteoarthritis; Line 158 “In this technique activation of channels”; minor typos along with the text);
  2. Figure 1: If possible, adjust graphic 1A yy axis so the significant differences at 4h can be better observed; Graphic B in figure caption is not correct, as it corresponds to figure 1A; The information “50 μl of 2% Carrageenan (100 mg, 5 ml saline)” and “respectively, after intra-plantar injections of carrageenan in the left 78 hind paw” should be placed in the methods section and possibly removed from the caption. In graph 1C, MPO meaning is not present, and the dispersion of the individual results appears to suggest a higher S.D., the authors choose to present the S.E.M. for a specific reason?
  3. The authors present the results in Figures 1A and 1E at 0h, 2h, and 4h, but 1D at 1h and 3h. Why did the authors evaluate the different parameters at different times?
  4. Figure 2, 3, and 5, please revise the captions accordingly.
  5. The authors have correctly addressed the EA toxicity issue in the discussion section. However, no data was presented regarding EA-induced toxicity in mice and cell culture. Have the authors analyzed any biomarkers in the mice used or cell viability assays in the primary cell culture that support the non-toxic concentrations used?
  6. Regarding cell culture size-frequency, the authors could refer in the caption the meaning of the “total” bar. Additionally, the significance of this assay is not extensively addressed. What was the contribution of this assay to the overall work?
  7. Line 345, please add HTAB full description since it is the first time in the text.

The discussion and materials & methods sections are well written, with detailed information regarding the topic here presented and the methodologies used.

Author Response

The article titled “(-)-Englerin-A has analgesic and anti-inflammatory effects in-2 dependent of TRPC4 and 5” further deepens the knowledge on this compound's anti-inflammatory and analgesic mechanism of action. The research was well planned, and the methods are well described.

In order to improve the manuscript, the authors should address some critical points, mainly in the results section.

  1. English language revision and spell check are recommended (e.g. “in vitro” in italic; in the abstract, EA is a compound extracted from a plant, not an extract, please revise; TRPC description, in the literature, is found “transient receptor potential cation channel”; Avoid the use of the same expression repeatedly, e.g. “a role for TRPC5 in an endogenous (…) in a model (…) in a model of osteoarthritis; Line 158 “In this technique activation of channels”; minor typos along with the text);

Reply: Thank you for your comment. The instances where in vitro and in vivo were not in italic were corrected (multiple instances, highlighted in yellow). We have checked the English and grammar, and the manuscript will be checked again by the editorial department, after it is accepted for publication.

According to the Oxford dictionary, an extract is “a substance that has been obtained from something else using a particular process. Eg. yeast extract”. Plant extract is a routinely used term and therefore we wish to keep it.

Transient receptor potential ion channels or TRPs are a large family of receptors comprising of 7 subfamilies: canonical (TRPC1-7), Melastatin (TRPM1-8), Vanilloid (TRPV1-6), Ankyrin (TRPA1), Mucolipin (TRPML1-3) and Polycystic (TRPP2,3 and 5) (DOI: 10.1111/bph.12532, DOI: 10.3390/cells9091983).

  1. Figure 1: If possible, adjust graphic 1A yy axis so the significant differences at 4h can be better observed; Graphic B of 2% Carrageenan (100 mg, 5 ml saline)” and “respectively, after intra-plantar injections of carrageenan in the left 78 hind paw” should be placed in the methods section and possibly removed from the caption. In graph 1C, MPO meaning is not present, and the dispersion of the individual results appears to suggest a higher S.D., the authors choose to present the S.E.M. for a specific reason?

Reply: Thank you. Fig 1A axis was corrected. The figure legends were corrected and “(100 mg, 5 ml saline)” and “in the left hidpaw” were removed and the information was added to the methods section: “Adult male mice were anaesthetized by 2% isoflurane carried in 0.5L/min O2 and paw oedema was induced by injecting 50 µl of 2% carrageenan (100 mg, 5 ml saline) into the pad region of the glabrous skin on the underside of the left hind paw of male mice using a 25G needle.” (line 325, highlighted in yellow). MPO meaning was added to the figure legends (multiple instances, highlighted in yellow). SEM was preferred to S.D. because it allowed for better visualisation of the time course data, which would otherwise be too crowded and imped proper interpretation of the data.

  1. The authors present the results in Figures 1A and 1E at 0h, 2h, and 4h, but 1D at 1h and 3h. Why did the authors evaluate the different parameters at different times?

Reply: In order to reduce the number of animals used, the thermal testing and mechanical testing were conducted in the same mice. This meant that we were unable to conduct the 2 assays simultaneously but separated them by one hour. This is in accordance with ethical guidelines for this project. The following sentence was added in the methods section (line 339, highlighted in yellow).” The 50% paw withdrawal threshold (PWT) was determined by increasing or decreasing stimulus intensity and estimated using the Dixon “up–down” method [27] at 2 and 4 hours post carrageenan injection to prevent overlap with thermal testing.”

  1. Figure 2, 3, and 5, please revise the captions accordingly.

Reply: Thank you. These have been revised (multiple instances).

  1. The authors have correctly addressed the EA toxicity issue in the discussion section. However, no data was presented regarding EA-induced toxicity in mice and cell culture. Have the authors analyzed any biomarkers in the mice used or cell viability assays in the primary cell culture that support the non-toxic concentrations used?

Reply: We regretfully did not consider looking at any toxicity biomarkers when we collected the data.

  1. Regarding cell culture size-frequency, the authors could refer in the caption the meaning of the “total” bar. Additionally, the significance of this assay is not extensively addressed. What was the contribution of this assay to the overall work?

Reply: The total refers to the size-distribution of the capsaicin responders, which helps validate the assay, as expectedly capsaicin responsive neurons are mostly small sized, but the description was missed in the figure legend. The following was added to the figure legend “total capsaicin responsive neurons (light grey, n=1453)” (Line 182, highlighted in yellow).

The assay was used as a more controlled setting to which test EA specificity over TRPC5, given that we know some neurons in the ganglia express TRPC5, if EA acted through TRPC5, we would expect to see no cobalt influx in the ganglia from KO mice. Therefore this assay reinforces our in vivo observation that EA does not act through TRPC5. The following is written in the discussion to clarify this: “To further confirm this, we also tested the ability of EA to induce cobalt uptake in cultured DRG neurons, which we have previously express TRPC5 channels [5]. In these cultured cells, EA induced cobalt influx in a dose dependent manner, similar to what was observed in TRPC5-transfected HEK cells [8], however, in cultured DRG neurons from TRPC5 KO mice the EA activity was surprisingly still present. Together our data demonstrates that both in vivo and in vitro effects of EA are not mediated solely by TRPC5 channels.” (line 277, highlighted in yellow)

  1. Line 345, please add HTAB full description since it is the first time in the text.

Reply: hexadecyltrimethylammoniumbromide replaced HTAB (line 359, highlighted in yellow) as it only appeared once in the text

The discussion and materials & methods sections are well written, with detailed information regarding the topic here presented and the methodologies used.

Round 2

Reviewer 3 Report

The authors have carefully revised the reviewers’ suggestions/corrections, which improved the overall quality and clarity of the manuscript.

The methodologies, captions, and additions to the results and discussion helped to clarify the experimental planning and therefore benefit the reader.

Regarding the cobalt uptake, the authors have explained why the use of this methodology. Although the main objective of the assay was to ascertain the EA-TRPC5 relation, the authors could improve the discussion section with an approach on which pathway could be involved, given the increased cobalt influx, and possibly correlate to the sentence in lines 309-312.

The author should revise for minor language revision and spell-check.

Author Response

The authors have carefully revised the reviewers’ suggestions/corrections, which improved the overall quality and clarity of the manuscript.

The methodologies, captions, and additions to the results and discussion helped to clarify the experimental planning and therefore benefit the reader.

Regarding the cobalt uptake, the authors have explained why the use of this methodology. Although the main objective of the assay was to ascertain the EA-TRPC5 relation, the authors could improve the discussion section with an approach on which pathway could be involved, given the increased cobalt influx, and possibly correlate to the sentence in lines 309-312.

The author should revise for minor language revision and spell-check.

Reply: Thank you for your comment that we agree improved the discussion of the cobalt uptake results. The following sentence was added to lines 284-289 : “One possible candidate mediating cobalt uptake in cultured DRG is TRPC4 which we have found, previously, to be expressed in DRG neurons [5]. Furthermore, selectivity as-says have demonstrated that EA can bind other TRP channels which are also expressed in DRG neurons [14, 16]. Future in vitro studies employing pharmacological tools or cells from transgenic animals could help elucidate which channel/s mediate the observed cobalt uptake in DRG.”